# Strong microwave squeezing above 1 Tesla and 1 Kelvin

Arjen Vaartjes[1,2], Anders Kringhøj [1,2], Wyatt Vine[1,2], Tom Day [1], Andrea Morello [1] & Jarryd J. Pla [1] ✉

Squeezed states of light have been used extensively to increase the precision of measurements, from the detection of gravitational waves to the search for dark matter. In the optical domain, high levels of vacuum noise squeezing are possible due to the availability of low loss optical components and high-performance squeezers. At microwave frequencies, however, limitations of the squeezing devices and the high insertion loss of microwave components make squeezing vacuum noise an exceptionally difficult task. Here we demonstrate direct measurements of high levels of microwave squeezing. We use an ultra-low loss setup and weakly-nonlinear kinetic inductance parametric amplifiers to squeeze microwave noise 7.8(2) dB below the vacuum level. The amplifiers exhibit a resilience to magnetic fields and permit the demonstration of large squeezing levels inside fields of up to 2 T. Finally, we exploit the high critical temperature of our amplifiers to squeeze a warm thermal environment, achieving vacuum level noise at a temperature of 1.8 K. These results enable experiments that combine squeezing with magnetic fields and permit quantum-limited microwave measurements at elevated temperatures, significantly reducing the complexity and cost of the cryogenic systems required for such experiments.

The measurement of weak signals is at the heart of many important challenges in modern science and engineering, from quantum computing[1], to spectroscopy[2], to the search for gravitational waves and dark matter[3,4]. Ultimately, the ability to measure a weak signal is constrained by noise, which at the quantum limit is dictated by vacuum fluctuations of the electromagnetic field. Vacuum fluctuations are a manifestation of the uncertainty principle, where the two quadrature components of a signal $I$ and $Q$ must obey the relation $\delta I^2 \delta Q^2 > 1/16$, where $\delta I^2$ ($\delta Q^2$) is the variance of the signal in the $I$ ($Q$) quadrature, measured in the unitless dimension of photons. This simple inequality provides a way to circumvent the seemingly fundamental constraint imposed by vacuum fluctuations: the noise along one quadrature can be reduced beneath the quantum limit, so long as the noise is increased in the opposite quadrature. This process is known as squeezing and can be exploited to enhance the signal-to-noise ratio (SNR) of a measurement.

A common approach to produce a squeezed state is to use a degenerate parametric amplifier (DPA) to amplify a vacuum state[5,6]. A DPA exhibits phase-sensitive gain, such that it amplifies noise along one quadrature and deamplifies (or squeezes) it along the other[7-10]. It is essential that the DPA behave as a noiseless (i.e. ideal) amplifier, as any noise added will degrade the squeezed state. Moreover, loss between the DPA and the detection apparatus also diminishes squeezing and must be minimized. These requirements make producing highly squeezed states an exceptionally difficult task.

In the optical domain, the availability of high-performance DPAs, ultra-low loss optical components and the relative ease of noiseless optical homodyne measurements have facilitated the demonstration of vacuum states squeezed by as much as 15 dB[11]. At microwave frequencies, Josephson junction-based parametric amplifiers represent the state-of-the art in microwave noise squeezers[4,12]. However, higher order nonlinearities present in Josephson parametric amplifiers (JPAs)

[1]School of Electrical Engineering and Telecommunications, UNSW Sydney, Sydney NSW 2052, Australia. [2]These authors contributed equally: Arjen Vaartjes, Anders Kringhøj, Wyatt Vine. ✉e-mail: jarryd@unsw.edu.au

constrain the amount of squeezing that can be achieved[6,8,13–15]. In addition, microwave components are lossy in comparison to their optical counterparts and microwave signals at the single photon level require near-noiseless amplification prior to measurement with conventional electronics[16]. As a result, many works have reported only an inferred measurement of squeezing[7,12,17], where degradation of the squeezed state between the DPA and detection system is not accounted for. Due to the additional complexity, there are few reports of direct measurements of microwave noise squeezing, where notably 4.5 dB of squeezing was utilized in an axion haloscope to speed up the search for dark matter[9].

Here we use a microwave squeezer made from a thin film of niobium titanium nitride (NbTiN), a material that exhibits a weak nonlinearity due to its kinetic inductance[10]. Unlike previous Josephson junction-based microwave squeezers, the kinetic inductance parametric amplifier (KIPA) is compatible with high magnetic fields and high temperatures and has negligible higher order nonlinearities[10]. We combine the KIPA with commercial microwave components and custom engineered device enclosures to minimize microwave losses, allowing us to achieve a direct measurement of − 7.8(2) dB of microwave vacuum noise squeezing. In addition, we show that high levels of squeezing can be maintained with the device operated in an in-plane magnetic field of up to 2 T, which permits its integration in applications such as spin resonance spectroscopy[8,18,19] and axion detection[4,9]. Finally, the high critical temperature ( ~ 13 K) of the superconducting film allows the squeezing of thermal states with the device operated at elevated temperatures, enabling vacuum level noise at a frequency of 6.2 GHz and a temperature of 1.8 K. These results show that quantum-noise-limited measurements can be performed at temperatures accessible using affordable pumped Helium-4 cryostats, a noise level that would otherwise only be attainable with comparatively expensive refrigeration technology.

## Results

### Squeezing setup

The KIPA devices exploit a weakly nonlinear kinetic inductance, which enables a three-wave mixing (3WM) interaction[10] in a half-wavelength coplanar waveguide (CPW) resonator (see Fig. 1a). Each KIPA is operated by applying a DC current $I_{DC}$ and a pump tone $I_p \sin(\omega_p t + 2\phi_p)$, with phase $2\phi_p$, at approximately twice the device's resonance frequency $\omega_p \approx 2\omega_0$. Here $\omega_0$ is the resonance frequency of the fundamental mode, with higher-order modes detuned away from multiples of $\omega_0$ by the kinetic inductance nonlinearity and dispersion[10,19]. The phase $\phi_p$ determines along which axes each KIPA squeezes or anti-squeezes. The DC current serves two purposes: it tunes the resonance frequency $\omega_0$ of the KIPA[20,21], and it strengthens the 3WM interaction, whilst leaving the Kerr strength negligibly low[22]. As a result, we strongly suppress higher order nonlinearities that have previously limited microwave squeezing experiments utilizing JPAs[6,13].

To facilitate direct measurements of microwave squeezing, we employ an experimental setup known as a squeezed state receiver (SSR)[9], with special care taken to minimize insertion loss and noise in the setup. In our experiments, the SSR is composed of two KIPAs with only a triple-junction circulator between them (Fig. 1b). To enable this configuration, we modify the KIPA design from those used in previous works[10,18] by adding a second port, allowing the pump and signal tones

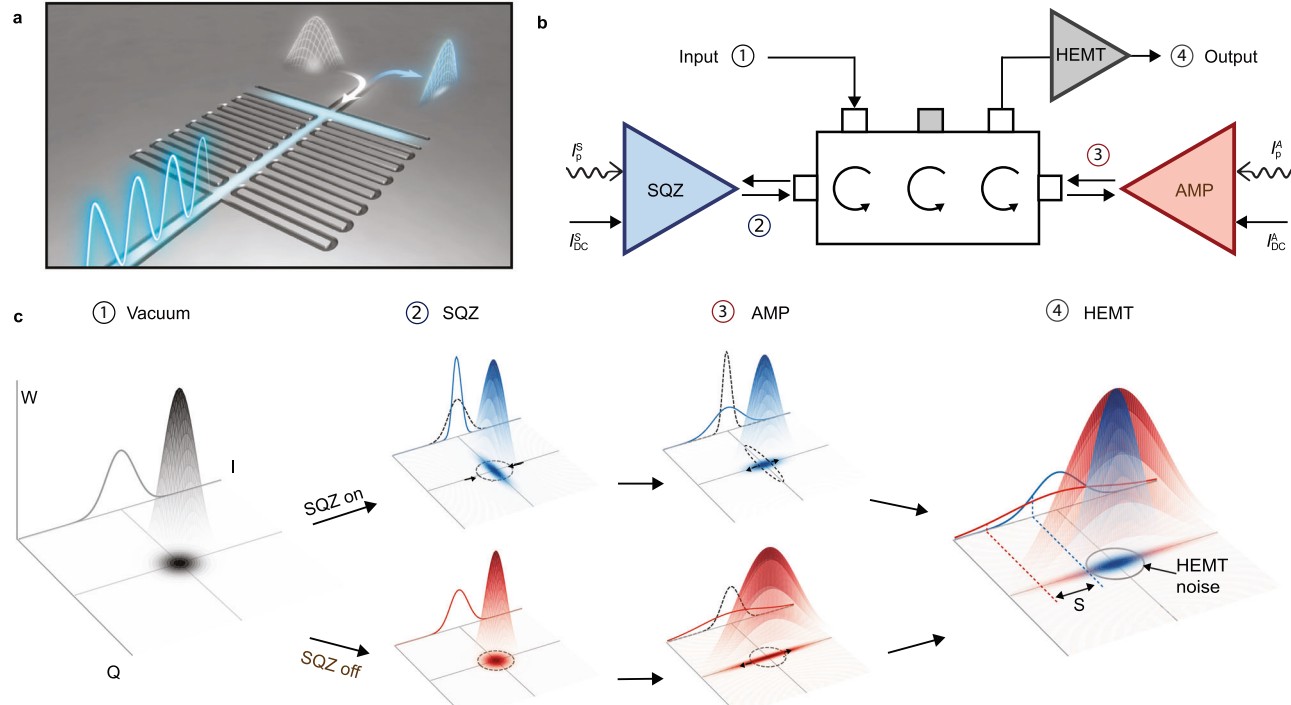

**Fig. 1 | Setup for a direct measurement of microwave vacuum squeezing.**
**a** Artist's impression of the squeezed state generation inside a two-port kinetic inductance parametric amplifier (KIPA). On the left, an AC pump tone (blue wave) and a DC current (blue glow) enter a half-wavelength coplanar waveguide (CPW) cavity with interdigitated capacitance. The DC current is shunted to ground via an inductance (blue glow). On the right side, a vacuum state (gray) gets reflected from the KIPA as a squeezed state (blue). **b** Two KIPAs are connected via a triple-junction circulator in a squeezed state receiver setup, with one labeled squeezer (SQZ) and the other labeled amplifier (AMP). The KIPAs are controlled with independent DC $I_{DC}$ and pump $I_p$ signals. **c** The procedure used to achieve a direct measurement of the squeezing produced by SQZ consists of two independent measurements. (1) A vacuum state is supplied to the input of the three-junction circulator and reflected off SQZ. (2) When SQZ is on (blue, top row) the state is squeezed along the $I$-quadrature. When SQZ is off (red, bottom row), it remains a vacuum state. (3) AMP is used to anti-squeeze (amplify) the resulting state along $I$, before the state is amplified by a high electron mobility transistor (HEMT) amplifier (4). The level of vacuum squeezing $S$ is defined by the difference (in decibels) of the noise power measured along $I$ at room temperature in the two scenarios (SQZ on and SQZ off). The 3D-plots represent the Wigner function $W$ of the various states.

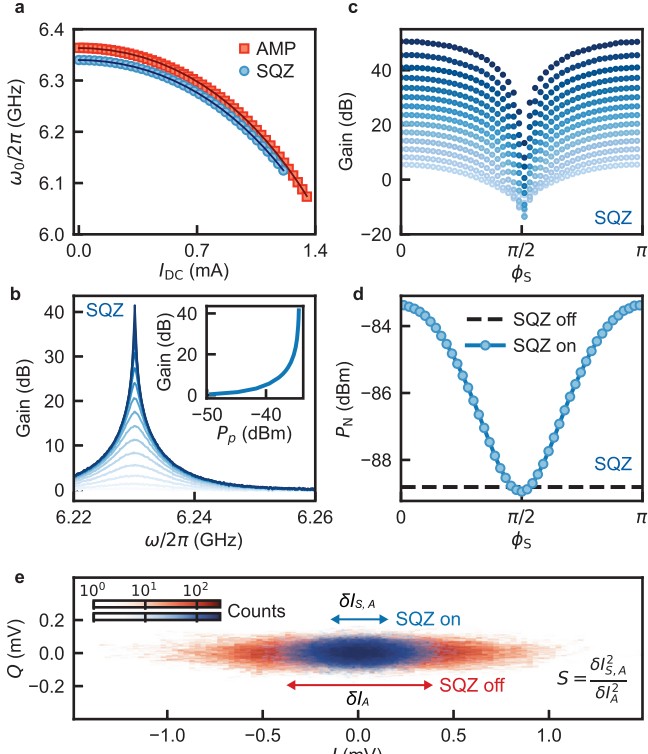

**Fig. 2 | Amplification and Squeezing with KIPAs. a** DC current $I_{DC}$ dependence of the resonance frequency $\omega_0$ of SQZ (blue) and AMP (red). Solid lines are fits to theory (see Supplementary Eq. S38). **b** Phase-insensitive gain of SQZ as a function of probe frequency $\omega/2\pi$ with a pump tone applied at $\omega_p/2\pi = 2\omega_0/2\pi = 12.46$ GHz and $I_{DC} = 0.90$ mA. Increasing opacity corresponds to increasing the pump power at the device input port from $P_p = -49$ to $-34$ dBm. Inset: Extracted maximum gain as a function of $P_p$. **c** Phase-sensitive gain at $\omega/2\pi = \omega_0/2\pi = 6.23$ GHz with $\omega_p = 2\omega_0$ for varying SQZ pump phase $\phi_S$. Increasing opacity corresponds to increasing the pump power from $P_p = -49$ dBm to $P_p = -33$ dBm. **d** Comparison of noise power $P_N$ for an input vacuum state with SQZ on (blue data points) compared to SQZ off (black dashed line) as a function of $\phi_S$. For $\phi_S = \pi/2$, maximum deamplification occurs, resulting in a noise level 0.14 dB below the reference system noise, which is measured with SQZ turned off. AMP is off for both measurements. **e** Histograms of the $I$ and $Q$ quadrature signals measured at $\omega/2\pi = 6.23$ GHz with only AMP on (red data points) and with both SQZ and AMP on (blue data points). The blue and red arrows show the standard deviations of the distributions along the $I$-quadrature: $\delta I_{S,A}$ and $\delta I_A$, respectively. The ratio of the variances corresponds to a direct measure of the achieved squeezing: $S = \delta I_{S,A}^2/\delta I_A^2$.

to be routed independently. We also make use of on-chip (superconducting) filters to suppress pump leakage (see Supplementary Note 3.B for further details). The two KIPAs are nominally identical but serve two distinct purposes in our experiments: we label the first SQZ because it is used to produce squeezed states from vacuum; we label the second AMP because it is used to noiselessly amplify the squeezed states. Both devices are installed in a dilution refrigerator with a base temperature of 10 mK.

We measure squeezing with a two-step protocol, displayed in Fig. 1c. In the first step SQZ and AMP are both activated, but are made to amplify along orthogonal quadratures, i.e., SQZ produces a squeezed vacuum state aligned on the $I$-quadrature, whereas AMP antisqueezes along $I$. We then measure the variance on $I, \delta I_{S,A}^2$. In the second step, we deactivate SQZ and make a measurement of the variance on same quadrature $\delta I_A^2$. Since the only difference between the two steps is the squeezing of the vacuum state, the vacuum squeezing level can be found from the ratio of the two variances $S = \delta I_{S,A}^2/\delta I_A^2$.

## Operation of the parametric amplifiers

Essential to the measurement protocol is that SQZ and AMP operate at the same frequency. This can be achieved over a 215 MHz span by using DC currents $I_{DC}$ to tune the two devices to a mutual resonance frequency $\omega_0$ (Fig. 2a). At $\omega_0/2\pi = 6.23$ GHz we measure the phase-insensitive gain of each KIPA by supplying a pump tone with frequency $\omega_p = 2\omega_0$ and varying the frequency $\omega$ of a probe tone. In Fig. 2b we show the measurement for SQZ where a maximum power gain of 41.5 dB was achieved for a pump power of $P_p = -33$ dBm, with a gain bandwidth product of $2\pi \times 17$ MHz. For phase-sensitive amplification, relevant for squeezing, we amplify coherent states with frequency $\omega_0$ while keeping the frequency of the pump tone fixed at $\omega_p = 2\omega_0$. Figure 2c shows that the gain of SQZ varies between 50 dB and $-14$ dB as a function of the pump phase of SQZ $\phi_S$. The performance of AMP is found to be nearly identical to SQZ (see Supplementary Note 4.C).

In the absence of a probe signal, the field input to SQZ is a vacuum state. By measuring the phase-dependent noise power $P_N$ along the $I$-quadrature, which is directly proportional to the variance of the noise, we demonstrate the squeezing capability of SQZ, with AMP deactivated. When $\phi_S \approx \pi/2$, the noise level (blue curve in Fig. 2d) falls slightly below the reference system noise, which is measured when both SQZ and AMP are deactivated (black dashed line in Fig. 2d). Notably, the amount of squeezing observed is modest ($-0.14$ dB). This is because the next activated amplifier in the detection chain – a high electron mobility transistor (HEMT) amplifier – adds 6.9 photons of noise to each quadrature (see Supplementary Note 4.J), overwhelming the squeezed noise. Even if the following amplifier were quantum-limited but operated in phase-insensitive mode (where a minimum of 0.25 photons of noise is added to each quadrature[23]), the maximum amount of squeezing that could be achieved is -3 dB. This emphasizes the need for a second DPA in direct measurements of vacuum squeezing, because it can boost the power of the squeezed state above the noise added by the HEMT without introducing additional noise.

Figure 2e presents a direct measurement of squeezing utilizing both SQZ and AMP, as outlined in the protocol described above. The measured noise is plotted as histograms in the $IQ$-plane, constructed from time traces of the noise (see Methods). The pump phases of SQZ and AMP are set to $\phi_S = \pi/2$ and $\phi_A = 0$ so that they maximally squeeze and amplify along $I$, respectively. When SQZ is deactivated (Fig. 2e, red data points), the vacuum state is amplified by AMP, resulting in an ellipse whose major axis is aligned along $I$. When SQZ is activated (Fig. 2e, blue data points), the variance along $I$ shrinks by an amount corresponding to the achieved vacuum squeezing level, $S$.

## Optimization of direct squeezing

To determine the maximum achievable squeezing in our system, we require the pump phase of SQZ $\phi_S$ and the pump phase of AMP $\phi_A$ to be aligned precisely, such that the two devices amplify along orthogonal axes. As expected, we observe high levels of squeezing whenever $\phi_S \approx \phi_A + \pi/2$ (Fig. 3a). At the optimal setpoint, we observe a maximum squeezing of $10 \log(S) = -7.8(2)$ dB (Fig. 3b, c). We emphasize that this level of squeezing is not an inferred value, but rather a direct measurement of an itinerant squeezed state without correcting for loss or system noise. The maximum squeezing presented in Fig. 3c is achieved at $\omega_0/2\pi = 6.23$ GHz, corresponding to $I_{DC} = 0.90$ mA for SQZ and $I_{DC} = 0.97$ mA for AMP. However, through the DC current tunability of the devices (Fig. 2a), we are able to demonstrate squeezing levels beyond $-7$ dB over a frequency span exceeding 100 MHz, see Supplementary Fig. S17a.

Squeezing is found to saturate with the gain of both amplifiers (Fig. 3d, e). This can be explained using a model based on input-output theory which captures several distinct mechanisms that limit $S$ (see Supplementary Note 2.B). In the high AMP gain limit, the squeezing

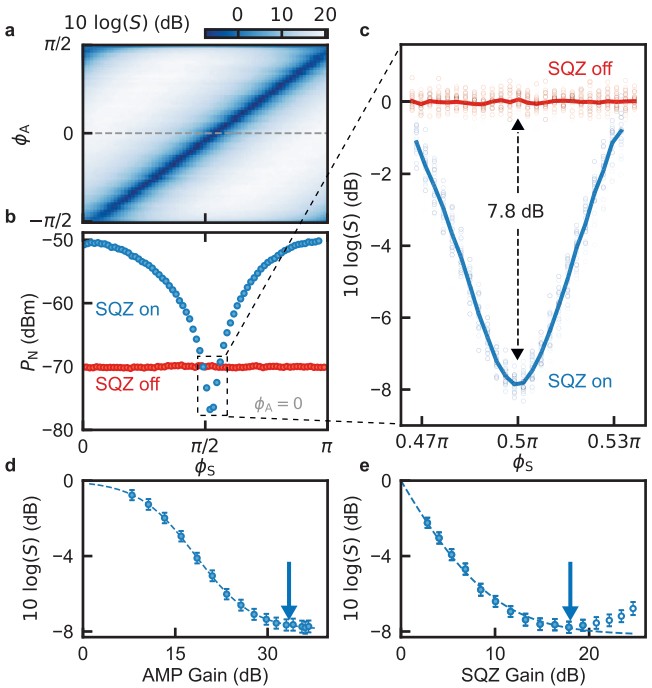

**Fig. 3 | Squeezing optimization. a** Squeezing $S$ as a function of SQZ pump phase $\phi_S$ and AMP pump phase $\phi_A$. The diagonal feature corresponds to SQZ and AMP squeezing along orthogonal axes, where $S$ is optimal. **b** A linecut at $\phi_A = 0$ in (**a**), showing the noise power $P_N$ on the $I$-quadrature in two scenarios: `SQZ on' and `SQZ off' (AMP is always on). Around $\phi_S = \pi/2$, SQZ reduces the noise on $I$ below the reference vacuum level. **c** A zoom-in around $\phi_S = \pi/2$, where SQZ is maximally squeezing along the $I$-quadrature. The squeezed noise reaches a minimum of 7.8(2) dB below the vacuum level. **d**, **e** Maximum squeezing as a function of AMP and SQZ gain. The gain of both amplifiers was measured via the anti-squeezing of a coherent state. Arrows indicate the setpoints used for the other panels. The dashed lines are fits to a cavity input-output model, which is derived in the Supplementary Information. The un-filled data points in **e** deviate from the model and are excluded from the fit. $\omega_0/2\pi = 6.23$ GHz for AMP and SQZ in all panels with probe frequency $\omega = \omega_0$. Error bars in (**d**) and (**e**) indicate the error on the mean.

level becomes

$$S = 1 - \frac{\eta_1(1 - G_S)(1/4 - n_S^{sq})}{1/4 + n_A^{anti}}, \qquad (1)$$

where $n_S^{sq}$ is the number of noise photons added by SQZ to $I$ when it is squeezing, $n_A^{anti}$ is the number of noise photons added by AMP to $I$ when it is anti-squeezing, $G_S$ is the squeezing gain ($0 \leq G_S \leq 1$) and $\eta_1$ is the SQZ to AMP transmission efficiency. Fitting our measurements to the full model (Supplementary Eq. S21) yields a combined factor $\eta_1(1/4 - n_S^{sq})/(1/4 + n_A^{anti}) = 0.85$. From independent measurements of the amplifier added noise we determine $n_S^{sq} = 0.02(2)$ and $n_A^{anti} = 0.00(2)$ (see Supplementary Note 4.J), which yields $\eta_1 = 0.92(8)$, equivalent to $-0.34$ dB of insertion loss. This finding is in close agreement with our estimates based on the components in our setup, as detailed in the Supplementary Information (Supplementary Note 1.B).

What is not captured by the model is the degradation of $S$ observed when the gain of SQZ exceeds 20 dB, which may be explained by an increased sensitivity of the deamplification gain to small drifts in $\phi_S$ as the pump power is increased (see Fig. 2c and Supplementary Fig. S18).

## Squeezing in magnetic fields

Several applications requiring strong magnetic fields benefit from the use of squeezed states, including electron spin resonance

spectroscopy[8,24] and dark matter axion detection[4]. Previous works have utilized JPAs but have required the amplifiers to be magnetically shielded. In contrast, several recent works have demonstrated magnetic field-resilient resonant amplifiers based on materials with kinetic inductance[25,26], though it remains an outstanding goal to realize highly squeezed states in such an environment.

To demonstrate the magnetic field-resilience of our squeezing setup, we modify our experiment so that SQZ is exposed to a magnetic field $B_\parallel$, while the rest of the components (including AMP and the circulator) are magnetically shielded (Fig. 4a). $B_\parallel$ is aligned to be in-plane with SQZ, to minimize the creation of magnetic flux vortices, which is expected to introduce additional loss in the resonator[27]. Figure 4b and c show that the squeezing level remains below $-7$ dB up to $B_\parallel = 1.5$ T. This level of squeezing is slightly less than was demonstrated in Fig. 3c, which might be explained by a slight shift of the optimal operation point in the modified setup.

The degree of squeezing we achieve is initially found to increase with $B_\parallel$, which is paired with an increase in $Q_i$ (see Supplementary Fig. S12). A similar effect was observed in ref. 28 for NbTiN resonators measured in in-plane magnetic fields. As $B_\parallel$ is raised beyond 1 T, however, $S$ becomes limited by the internal quality factors of SQZ and AMP, both of which begin to degrade. Interestingly, the maximum field at which we measured squeezing (2 T) is limited by AMP, not SQZ. At larger fields, AMP loses superconductivity as a result of the stray magnetic field, which is a factor of ~ 200 less than $B_\parallel$, but is not aligned in-plane. This implies that a high degree of squeezing can be achieved at fields greater than 2 T with careful alignment of the magnetic field with both amplifiers.

## Squeezing at elevated temperatures

Next, we leverage the high critical temperature of our amplifiers to demonstrate squeezing of the noise produced by a hot thermal bath down to the vacuum level. By heating up not only the input state, but also SQZ and the circulator, we demonstrate that our setup is capable of reducing thermal microwave noise to the quantum limit, at temperatures up to 1.8 K.

We detect the squeezed thermal states with AMP, which is mounted on the mixing chamber of the dilution refrigerator and remains at its base temperature $T_{MC} \approx 10$ mK. In contrast, SQZ and the circulator are both thermally anchored to a variable thermal source (VTS) which is heated to a temperature $T_H$ (Fig. 4d). Raising $T_H$ generates a thermal state containing $n$ photons that is directed to the input of SQZ. The photon number per quadrature is given by $n = n_{th} + 1/4$, where $n_{th} = [\exp(\hbar\omega/k_B T_H) - 1]^{-1}/2$ is the number of thermal photons in each quadrature, with $k_B$ the Boltzmann constant and $\hbar$ the reduced Planck constant.

In Fig. 4e, we present measurements performed at four values of $T_H$. At each setpoint, we measure the noise power of the signal with both amplifiers activated ($P_N^{S,A}$) and when only AMP is activated ($P_N^A$). The latter is indicated in Fig. 4e as a series of dashed lines, which rise in power with $T_H$ because of the increased thermal noise $n(T_H)$ as the input state is heated. At each temperature, $P_N^{S,A}$ can be reduced below $P_N^A$ for pump phases $\phi_S \approx \pi/2$. We define this reduction to be the level of thermal squeezing, $S_{th}$. In Fig. 4f we show that $S_{th}$ initially increases with $T_H$, and saturates at $10\log(S_{th}) = -10.2(1)$ dB for $T_H$ in the range $1 - 2$ K. As $T_H$ is increased further, $S_{th}$ begins to deteriorate, which we attribute to a decline in the internal quality factor $Q_i$ of SQZ with $T_H$ (see Supplementary Fig. S13).

We note that the maximum level of thermal squeezing achieved here is greater than the vacuum squeezing shown in Fig. 3c. This is because as the temperature increases, the impact of KIPA-added noise on squeezing reduces (see Supplementary Note 2.D), in which case the squeezing is predominantly limited by the insertion loss between SQZ and AMP. This allows us to estimate the transmission efficiency between SQZ and AMP as $\eta_1 = 0.927(3)$ for this experiment, which

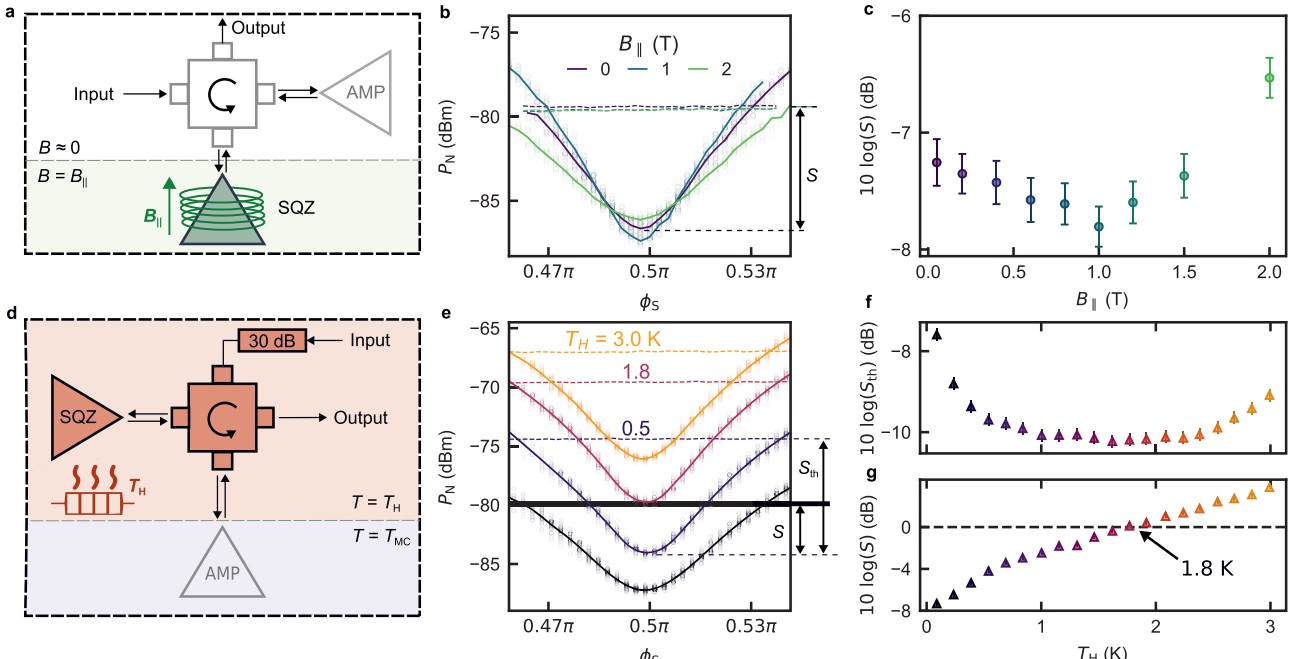

**Fig. 4 | Magnetic field resilience and thermal squeezing. a** Modified squeezing setup, where SQZ is placed in an in-plane magnetic field $B_\parallel$. All other components are magnetically shielded such that $B \approx 0$ T. **b** Noise reduction with respect to the reference vacuum level (dashed lines) for fields up to $B_\parallel = 2$ T. **c** Vacuum squeezing as a function of magnetic field $B_\parallel$. The squeezing level remains below − 6.3 dB up to 2 T. **d** Modified setup for thermal squeezing. The circulator and SQZ are thermally anchored to a heat source with variable temperature $T_H$. AMP is thermally isolated at $T = T_{MC} \approx 10$ mK. A 30 dB input attenuator is also heated to $T_H$, thermalizing the input state to this temperature. **e** Noise squeezing for several values of $T_H$. The horizontal dashed lines indicate a rise in noise power due to the elevated temperature. The second line from the top (pink) shows that thermal noise at $T_H = 1.8$ K

is squeezed down to the vacuum level (black horizontal solid line). The vertical arrows indicate the definitions of vacuum squeezing $S$ and thermal squeezing $S_{th}$. **f** Squeezing $S_{th}$ with respect to a thermal input state. Note that the thermal squeezing level reaches − 10 dB; in this configuration the squeezing is primarily limited by the insertion loss between SQZ and AMP. Above $T_H \approx 2.25$ K, $S_{th}$ begins to degrade due to a reduction in the internal quality factor of SQZ (see Supplementary Note 4.I). **g** Squeezing with respect to the vacuum $S$ as a function of heater temperature $T_H$. At $T_H = 1.8$ K, the squeezed noise power is equal to the power of the vacuum fluctuations. Error bars in **c** and **f** indicate the error on the mean. The error bars in (**g**) are smaller than the markers.

translates to an insertion loss of − 0.33 dB (see Supplementary Fig. S19). Although the experimental setup depicted in Fig. 4d differs from the one in Fig. 1b in that it includes an additional superconducting coaxial cable between SQZ and AMP, the extracted insertion loss is in close agreement with the earlier found value of − 0.34 dB.

The squeezed thermal states can also be measured relative to the vacuum level (black horizontal line in Fig. 4e). By comparing the measurements of $P_N^{S,A}$ taken at temperature $T_H$ with the vacuum reference (see Supplementary Fig. S20), we show that $S$ degrades monotonically with $T_H$ (Fig. 4g). At $T_H = 1.8$ K, the level of squeezing is $10 \log(S) = 0$ dB, indicating that the noise of the thermal state has been squeezed to the vacuum level.

## Discussion

There are several avenues available for improving the amount of vacuum squeezing achieved. A significant limitation is the insertion loss of the microwave components that the itinerant squeezed state traverses[9]. The data in Fig. 4g suggests up to −10 dB of squeezing could be reached with the existing setup if the insertion loss were the only factor degrading squeezing. Whilst already low, the − 0.34 dB of insertion loss could be reduced further by integrating the squeezing devices within the same enclosure as the circulator, circumventing the need for connectors along the squeezed state path, or fully integrating the two KIPAs using on-chip circulators[29,30].

A second factor that limits vacuum squeezing is the noise added by the two KIPAs. Given the measured noise performance of SQZ and AMP, squeezing is limited to − 11.0 dB in the absence of insertion loss (Eq. (1)). This indicates that insertion loss and added noise combine almost equally to limit squeezing in our experiment. The KIPA added

noise could be reduced further by carefully impedance-matching the pump port and the cavity or by reducing the impedance of the cavity, thereby lowering the required pump power and any associated heating.

Phase instability in the setup can also limit squeezing. Any deviation from the optimal squeezing angle $\phi_S = \pi/2$ will mix in gain-dependent anti-squeezed noise, with this penalty becoming increasingly severe as the squeezer gain is increased. This is a known source of squeezing degradation in optical setups and techniques that improve phase stability (e.g. phase-locked loops) could also be employed in future microwave squeezing experiments[31].

Besides the ease of fabrication, the single NbTiN layer design of these devices give the squeezers a unique resilience to magnetic fields and elevated temperatures, making their use attractive in a variety of applications. For instance, the high levels of microwave squeezing already demonstrated here inside magnetic fields and at high temperatures could find immediate application in electron spin resonance (ESR) spectroscopy[8,18], permitting quantum-limited spin detection at temperatures consistent with conventional ESR spectroscopy operating conditions ($\gtrsim 2$ K), but without the need for expensive cryogenic systems. Similarly, the search for dark matter axions[4,9] could be sped up by a factor of six compared to the quantum limit, or axion detectors could be simplified (negating the need for extensive magnetic field compensation and shielding) and made more widely available by installing them in low-tech pumped Helium-4 cryostats.

Squeezed vacuum states are also a valuable resource in measurement-based computation schemes using entangled cluster states encoded in the modes of an electromagnetic field[32,33]. Critically, it has been shown that fault-tolerance with this approach can be

attained using vacuum states squeezed by at least 10 dB[34], which is within reach of our current setup. Superconducting circuits also offer the ability to deterministically prepare non-Gaussian states of light[35], an essential resource in such schemes.

## Methods

### KIPA Hamiltonian

The Hamiltonian governing the operation of the KIPA is given by

$$H_{\text{KIPA}}/\hbar = \Delta \hat{a}^\dagger \hat{a} + \underbrace{\frac{\xi}{2}\hat{a}^{\dagger 2} + \frac{\xi^*}{2}\hat{a}^2}_{\text{3WM}} + \underbrace{\frac{K}{2}\hat{a}^{\dagger 2}\hat{a}^2}_{\text{4WM}},\qquad(2)$$

which is defined in a frame rotating at half of the pump frequency $\omega_p/2$[10]. Here $\hat{a}$ ($\hat{a}^\dagger$) is the bosonic annihilation (creation) operator, $\Delta = \omega_0 - \omega_p/2$ is the frequency detuning, $\xi = -e^{-i2\phi_p}\omega_0 I_{\text{DC}} I_p/(4I_*^2)$ is the 3WM strength, where $I_*$ determines the susceptibility to a DC current. $K$ is the self-Kerr interaction responsible for four wave mixing (4WM). The ratio $|\xi/K|$, which is used to indicate how susceptible a 3WM device is to higher-order nonlinearities[13], can be as large as $10^{8}$ [10].

### Device fabrication

The devices are fabricated on 15 nm thick films of NbTiN deposited on an intrinsic, high-resistivity silicon chip. The devices are patterned using electron beam lithography followed by a reactive ion etch with an $CF_4$:Ar plasma. The devices both consist of four main components: the first is a band-stop stepped-impedance filter constructed from a series of nine quarter-wavelength ($\lambda/4$) CPW segments with alternating high- ($Z_{\text{hi}}$) and low-impedance ($Z_{\text{lo}}$), through which we introduce the DC current $I_{\text{DC}}$ and pump $I_p$. The second component is a half-wavelength ($\lambda/2$) resonator, based on an interdigitated capacitor design. The third component is a fifth-order low-pass Chebyshev stepped impedance filter, which is constructed from five alternating $Z_{\text{hi}}$ and $Z_{\text{lo}}$ segments of varying lengths, through which we introduce the signal. The on-chip low-pass filter is crucial to avoid KIPA pump tone cross-talk. The final component is a shunt inductance to ground, which dictates the coupling quality factor $Q_c$ of the resonator. The inductance is chosen so that $Q_c \approx 200$, thereby ensuring the resonator is strongly over-coupled to the signal port. See Supplementary Note 3.B for further details on the device design and simulations.

### Device packaging

The two KIPA devices were mounted to their own enclosures. Both enclosures were constructed from gold-plated oxygen-free copper and act as a rectangular waveguide whose cutoff frequency is greater than the KIPAs, thereby suppressing radiation losses. The two ports of each KIPA were wire bonded to minimally-sized printed circuit boards, to minimize dielectric and ohmic losses. The devices were connected in series via a low-loss triple junction circulator. All measurements were completed with the devices mounted to the mixing chamber of a dilution refrigerator with a base temperature of 10 mK. See Supplementary Note 1 for further details on the device enclosure, mounting, and fridge wiring.

### Homodyne reflection measurements

Measurements of the devices are performed in reflection and recorded with a vector network analyzer (VNA), a spectrum analyzer or an oscilloscope.

We use a VNA to measure the resonance frequency and quality factors of the resonators, which we obtain from fits of $S_{11}(\omega)$ to a cavity input-output model (see Supplementary Note 4.A). The phase-insensitive gain of both SQZ and AMP are acquired with the same setup.

We acquire the noise ellipses in Fig. 2e by demodulating the output noise with an *IQ*-mixer and recording the *I* and *Q* time traces on an oscilloscope with a sampling rate of 8.33 MSa/s. We apply a digital low-pass filter with a cutoff frequency of 150 kHz (the same cutoff frequency as in the squeezing measurements of Figs. 3, 4) to filter out noise outside of the bandwidth of SQZ and AMP. We then bin the time traces to create a histogram in the *IQ*-plane.

The noise squeezing measurements are performed with a spectrum analyzer by measuring the noise power $P_N$ of one quadrature of the homodyne-demodulated output signal. We measure the demodulated noise on *I* with a bandwidth resolution of 300 Hz and average over a bandwidth of 120 kHz (30 kHz-150 kHz, see Supplementary Fig. S17b for the bandwidth dependence of squeezing). An empirical definition of squeezing is $S = P_N^{S,A}/P_N^A$ where $P_N^{S,A}$ and $P_N^A$ are the noise powers measured when both SQZ and AMP are activated and when only AMP is activated, respectively.

We find the maximum squeezing level by applying a simple protocol. First, we set $\phi_A = 0$ to maximize $P_N^A$. We then turn on SQZ and measure $P_N^{S,A}(\phi_S)$ to estimate the $\phi_S$ that minimizes $S$. We take repeated measurements across a small range of $\phi_S$ and record $S(\phi_S)$ using interleaved measurements of $P_N^A$ and $P_N^{S,A}$. We account for phase drifts in the microwave pump sources by an alignment procedure displayed in Supplementary Fig. S18.

### Measurements in magnetic fields

For the experiments where SQZ is measured in a magnetic field, a 6 T/1 T/1 T vector magnet is installed in the dilution refrigerator. SQZ is placed in the center of the magnet, on the cold finger of the dilution refrigerator and connected to the circulator via a short length of semi-rigid NbTi superconducting cable, with the setup otherwise identical compared to the experiments performed at zero field. AMP and the circulator, situated on the mixing chamber plate, are partially magnetically shielded by a lead enclosure.

### Measurements at elevated temperatures

For the experiments where SQZ is used to squeeze thermal states, SQZ and the circulator are thermally anchored to the thermal noise source, while AMP is thermally anchored to the mixing chamber plate and connected to the circulator via a a short length of semi-rigid NbTi superconducting cable.

## Data availability

All data needed to evaluate the conclusions in the paper are present in the paper and/or the Supplementary Information. The raw data from this study are openly available in the Zenodo repository https://zenodo.org/uploads/10730079.

## Code availability

Measurement scripts, analysis code used to process the raw data and scripts to create all figures are available in the online repository https://zenodo.org/uploads/10730079.

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

## Acknowledgements

J.J.P. acknowledges support from an Australian Research Council Discovery Early Career Research Award (DE190101397). J.J.P. and A.M. acknowledge support from the Australian Research Council Discovery Program (DP210103769). A.M. is supported by the Australian Department of Industry, Innovation and Science (Grant No. AUS-MURI000002). A.K. acknowledges support from the Carlsberg Foundation. A.V., W.V. and T.D. acknowledge financial support from Sydney Quantum Academy, Sydney, NSW, Australia. This research has been supported by an Australian Government Research Training Program (RTP) Scholarship. The authors acknowledge support from the NSW Node of the Australian National Fabrication Facility. We thank Robin Cantor and STAR Cryoelectronics for sputtering the NbTiN film, and David Niepce from Low Noise Factory for discussions on circulator losses. Finally, we acknowledge Tony Melov for producing the artist's impression of the device in Fig. 1a.

## Author contributions

A.V., A.K., and W.V. performed the experiments and analyzed the data. A.K., W.V. and J.J.P. designed the device. W.V. fabricated the devices. T.D. designed the device enclosures and variable thermal source. J.J.P. and A.M. supervised the project. A.V., A.K., W.V., and J.J.P. wrote the manuscript with input from all authors.

## Competing interests

J.J.P. is an inventor on a patent related to this work (AU2020347099) filed by the University of New South Wales with a priority date of 09 September 2019. The authors declare that they have no other competing interests.

## Additional information

**Peer review information** : *Nature Communications* thanks Baleegh Abdo, Roberto Di Candia and Mingrui Xu for their contribution to the peer review of this work. A peer review file is available.

