## [Peer Review File · Nature Communications]

Strong Microwave Squeezing Above 1 Tesla and 1 KelvinREVIEWER COMMENTS

Reviewer #1 (Remarks to the Author):

The manuscript presents an experimental implementation of a direct measurement of a squeezed vacuum/thermal state, achieving a record of 7.8 dB below-vacuum using a Kinetic Inductance Parametric Amplifier (KIPA). Previous direct squeezing detection was reported in [2], amounting to 4.5 dB below-vacuum. This achievement remains robust within a wide, yet improvable, range of magnetic fields applied to the generating KIPA (up to 2T) and anchored temperatures (up to 1.8K).

I find the paper intriguing as achieving high squeezing in a strong magnetic field has direct applications in relevant quantum sensing scenarios, such as axion detection. Moreover, operating at a temperature above 1K notably reduces the cryogenic cost of the experiment. The author also claims that the proposed setup can be further optimized, with a potential for achieving 10 dB squeezing below-vacuum under the same conditions. This is a crucial element for fault-tolerant quantum computing using GKP states.

Having said that, I suggest publishing this paper in Nature Communications.

I kindly ask the authors to address the following minor points:

1. Page 1, column 1. "A common approach to produce... a vacuum state [6]". I would cite APL 93, 042510 (2008) here.
2. Is there a specific reason to work at the point $I_{\text{DC}}=0.9$ mA?
3. The pump phases ϕ_S and ϕ_A are not defined in the text. I suggest defining them, perhaps when introducing ϕ_p .
4. Concerning the measurement of the squeezing level with AMP off, i.e., Fig. 2d. I noticed that the HEMT noise has been characterized. Has the HEMT gain been characterized as well? If so, one can use the data from Fig. 2d for the indirect measurement of squeezing at the

HEMT point by (theoretically) reversing the HEMT channel. This shall be compared with the value obtained with direct measurement, i.e., using AMP.

5. Page 4, below Eq. 1. It would be better to specify that η_1 is the SQZ to AMP transmission efficiency.

6. Eq. 2 of Suppl. Mat.: What are γ and κ ?

7. In Eq. 6 of Suppl. Mat. and in the whole text: What does the subscript k stand for?

8. Below Eq. 7, $G_{k-1} < 1$ should be $G_{k-1} < 0$.

Reviewer #2 (Remarks to the Author):

“Strong Microwave Squeezing Above 1 Tesla and 1 Kelvin” reports on the measurements of strong microwave squeezing ~ 8 dB in a resonator-based kinetic inductance parametric amplifier (KIPA) made of NbTiN, where squeezing refers to the quantum effect in which the noise variance of coherent states is reduced below the uncertainty limit in one quadrature of the electromagnetic field at the expense of increasing it in the other.

Owing to this unique property, which is closely related to the formation of entanglement between bosonic modes, squeezing of microwave vacuum or microwave signals can be useful in several applications in quantum sensing and quantum information processing.

The most common method for generating and studying squeezing of microwave signals is by parametrically modulating the nonlinear inductance of Josephson junctions (JJs) embedded in superconducting devices. While this method is very successful and reliable, it has several disadvantages, which this work tries to overcome. One is the requirement of JJs (Al/AlO/Al) to be cooled to temperatures much lower than 1 K (tens of mK), which typically necessitates the use of dilution fridges that are expensive. Two, is the requirement of JJs to be well shielded from magnetic fields to prevent degradation in their performance. While the first requirement can be relaxed using, for example, Nb-based JJs, the second is more difficult to

satisfy with devices that use JJs.

One high-profile application which could potentially benefit from using KIPAs, which exhibit strong microwave squeezing while being insensitive to in-plane magnetic fields, is axion (dark matter) search experiments, which require, for their detection, ultra-low microwave noise and the application of magnetic fields.

Other points of novelty of this work include the use of two stages of KIPA devices to perform direct measurement of squeezing (known as squeezed state receiver experiments in which one of the devices is operated as a squeezer while the other as a phase-sensitive amplifier), the deliberate effort to minimize the insertion loss between the two KIPA devices by mounting them on the same bracket separated only by a low-loss triple junction circulator, and the use of on-chip bandstop and lowpass filters, which allow the dc and microwave pump signals to be fed through a separate port from the one carrying the input and output microwave signals.

In general, the results of the paper are very interesting and important and suitable for a wide audience. The manuscript is well written. The figures are clear and illustrative. The authors have made a good effort to support their main claims and to justify their conclusions. They also included very detailed information about their measurements, setup, and characterizations in the supplementary information. But prior to recommending the publication of the manuscript in Nature communications, I would like to ask the authors to address a few comments and questions:

1- The photo of the actual device is very small and buried deep in the supplementary material. In my opinion, the device photo or circuit should be included in the main text (figure 1a is not representative) or at least it should be enlarged in the supplementary information and its various elements should be described in more detail in the text. Also, the circuit symbol of an inductor to ground is confusing. It could refer to the part of the device that implements an inductance to ground or indicate that there is an additional inductor to ground that is not shown. Maybe include an additional drawing which better represents the circuit elements of the layout in a more representative way than the

equivalent lumped parallel RLC drawn in c.

2- On several occasions, the authors refer to their resonator as a $\lambda/2$ resonator, is it really a transmission line resonator whose length is $\lambda/2$ (where λ is the resonance wavelength) or is it a lumped element device with a finite number of interdigitated fingers to ground extending from a lumped inductor in the form of a narrow superconducting wire?

3- Are there other resonance frequencies of this KIPA or higher harmonics? If yes, do they play a role in the device performance?

4- How was the inductor to ground that sets the coupling Q of the device determined? Through simulations, or measurements of different devices with different inductance values?

5- In several locations, the authors use the term nondegenerate gain or nondegenerate amplification when maybe the better term is phase-insensitive gain/amplification (since the KIPA has only one mode and one port).

6- Is the 20 dB attenuation of the lowpass filter at the pump frequency enough to block the pump from being transmitted through the device microwave port?

7- Are the pump powers cited in the caption of figure 2 correspond to the powers at the device port?

8- I might have missed it but where in the main text or supplementary information do you cite and discuss the extended data figures?

9- The gain-bandwidth of the device is quite small ~ 17 MHz, could you discuss in the paper how this figure can be increased and how that might affect the device performance?

10- Is the lowpass filter used at the mixing chamber a series RC like the one on the 4 K stage,

with the same R and C but with a copper wire? What is the R? Does it cause heating when the dc current is applied?

11- The labels d and e for figure 3 are far above their respective graphs.

12- Given the narrow bandwidth of the KIPAs is it difficult to align their resonance frequencies? How precise is the alignment and how long does it stay stable?

Baleegh Abdo

Reviewer #3 (Remarks to the Author):

In this paper, the authors thoroughly demonstrate the advantages of a novel 3WM kinetic-inductance parametric amplifier, by showing record level “directly-measured” microwave squeezing of 7.8 dB, resilience to environments with elevated temperature (1K), and external magnetic fields (1T). The record level of squeezing is made possible because kinetic inductance has relatively low spurious 4WM nonlinearity, which is typically a limiting factor of the level of squeezing achievable by a Josephson-junction based amplifier. Resilience to high temperature/magnetic field is made possible by the high band gap of the material of choice (NbTiN). Even though this is not the first time such an amplifier device has been reported, the characterizations that reveal unprecedented squeezing performance are new. The “Squeezes state receiver” setup is very thoughtfully designed and experiments are well executed. The work will inspire the community to explore the use of squeezing in applications that were not possible before. Therefore, it deserves to be published on Nature Communications, if the questions below are addressed.

Technical comments:

Validity of directly-measured squeezing level results:

For the comparison of P_N when squeezer is on vs off to be valid (fig 3b and 3c), an important underlying assumption needs to be true – the system gain (from the output of

the squeezer and the room temperature ADC) needs to be the same when the squeezer is turned on and off. While in reality, it is not trivial to maintain consistent KIPA amplifier gain when the input signal changes drastically. For example, by turning on the squeezer, it is certainly possible that the leaked squeezer pump slightly saturates the amplifier, and therefore reduces the gain of the amplifier. In that case, the squeezing level inferred from Figure 3c COULD BE EXAGGERATED. Can the author talk about what test they have done to rule out this possibility? (maybe from the view of 1dB compression point?) Results in supplemental material 5 bring some confidence, but are not sufficient. Evidence of consistent amplifier gain when SQZ is on vs off needs to be shown at all conditions where the squeezing level is measured.

Can the author talk more about the material and design of the magnetic shield used to protect AMP and circulator? Maybe adding a picture to figure S2 would be helpful. Shielding 2T of field for components that are within a few centimeters seems not a trivial task (distance between circulator and SQZ in figure S2a)

Figure 4d can be made more obvious that the input to the SQZ is also heated. Maybe include a drawing of the 30 dB attenuator at the “input” of the circulator?

Typo In supplemental material 3B: “The LP-SIF is introduced to filter out the pump tone at $\omega_p \approx 2\omega_0$ in order to limit crosstalk between SQZ and AMP (see Fig. S6d).” The authors may intend to refer to Fig S6e instead.

Response to Reviewers - Strong Microwave Squeezing Above 1 Tesla and 1 Kelvin

We would like to thank the Reviewers for their valuable comments. We have addressed all points raised by the Reviewers and modified the manuscript accordingly. We have submitted updated versions of the main text and the Supplementary Information. We also provide copies of the main text and Supplementary Information with our revisions highlighted in red for deleted text and in blue for added text. References to the main text in the below responses refer to the resubmitted manuscript.

The experimental data and the scripts used for analysis and figure generation have been uploaded to an open access repository which can be accessed via the following URL: <https://zenodo.org/uploads/10730079>.

RESPONSE TO REVIEWER 1

The manuscript presents an experimental implementation of a direct measurement of a squeezed vacuum/thermal state, achieving a record of 7.8 dB below-vacuum using a Kinetic Inductance Parametric Amplifier (KIPA). Previous direct squeezing detection was reported in [2], amounting to 4.5 dB below-vacuum. This achievement remains robust within a wide, yet improvable, range of magnetic fields applied to the generating KIPA (up to 2T) and anchored temperatures (up to 1.8K). I find the paper intriguing as achieving high squeezing in a strong magnetic field has direct applications in relevant quantum sensing scenarios, such as axion detection. Moreover, operating at a temperature above 1K notably reduces the cryogenic cost of the experiment. The author also claims that the proposed setup can be further optimized, with a potential for achieving 10 dB squeezing below-vacuum under the same conditions. This is a crucial element for fault-tolerant quantum computing using GKP states. Having said that, I suggest publishing this paper in Nature Communications.

We thank the Reviewer for their complimentary assessment of our results. In the following we address all points made by the Reviewer. The Reviewer's comments are highlighted in blue with our responses noted below in black:

1. Page 1, column 1. "A common approach to produce... a vacuum state [6]". I would cite APL 93, 042510 (2008) here.

We thank the Reviewer for suggesting this paper on microwave deamplification with a flux-driven JPA, which we now cite (new Ref. [11]).

2. Is there a specific reason to work at the point $I_{DC} = 0.9$ mA?

We operate at this current setpoint as this is where we achieve the maximum amount of vacuum squeezing. However, as shown in the Extended Data Fig. 5, we can achieve squeezing levels below -7 dB over a frequency span exceeding 100 MHz. We have now added an explicit explanation in Section IV of the main text (lines 200-206), where we also refer to the extended data Fig. 5:

The maximum squeezing presented in Fig. 3c is achieved at $\omega_0/2\pi = 6.23$ GHz, corresponding to $I_{DC} = 0.90$ mA for SQZ and $I_{DC} = 0.97$ mA for AMP. However, through the DC current tunability of the devices (Fig. 2a), we are able to demonstrate squeezing levels beyond -7 dB over a frequency span exceeding 100 MHz, see Extended Data Fig. 5a.

We have also mentioned the measurement frequency in the captions of Fig. 2e and Fig. 3e to make the operation point unambiguous.

3. The pump phases ϕ_S and ϕ_A are not defined in the text. I suggest defining them, perhaps when introducing ϕ_P .

We thank the Referee for pointing this omission out. We now define ϕ_S and ϕ_A the first time that they each appear in the main text.

4. Concerning the measurement of the squeezing level with AMP off, i.e., Fig. 2d. I noticed that the HEMT noise has been characterized. Has the HEMT gain been characterized as well? If so, one can use the data from Fig. 2d for the indirect measurement of squeezing at the HEMT point by (theoretically) reversing the HEMT channel. This shall be compared with the value obtained with direct measurement, i.e., using AMP.

The Reviewer makes an excellent suggestion regarding whether we could infer a squeezing value based on the measurement in Fig. 2d of the main text. As the Reviewer notes, we would require a calibration of the HEMT gain to do this. Measuring the HEMT gain would have required the use of a switch in the setup after the AMP

to bypass the HEMT. We opted against this approach due to concerns about increased insertion loss, which could have limited the squeezing.

With our present dataset, we do not have direct access to the HEMT gain G_H . The slope of the noise measurements in Supplementary Information Fig. S14d only gives us access to the product $\eta_0\eta_1\eta_2G_H$. Whilst we were able to extract η_1 through an analysis of our squeezing data (see main text Section IV and extended data Fig. 7), the uncertainty on the transmission parameters η_0 and η_2 prevents us from accurately inferring a squeezing value based on the data in Fig. 2d of the main text.

5. Page 4, below Eq. 1. It would be better to specify that η_1 is the SQZ to AMP transmission efficiency.

We agree with the Reviewer that this is a more appropriate definition. We have modified the sentence as follows (see lines 216-217):

“... and η_1 is the SQZ to AMP transmission efficiency”.

6. Eq. 2 of Suppl. Mat.: What are γ and κ ?

We thank the Referee for pointing this out. γ and κ are the internal and external rates of loss of the resonator, respectively. We have now added explicit definitions (see line 81 of the Supplementary Information).

7. In Eq. 6 of Suppl. Mat. and in the whole text: What does the subscript k stand for?

The subscript “k” represents KIPA in our work, and it is used in Supplementary Section II.A.1 for a general KIPA. In later sections, the subscript k is replaced by “S” or “A” to specify SQZ or AMP specifically. To enhance clarity, we have updated it to a capital “K” throughout the manuscript. Additionally, we have included an explicit reference to the meaning of the subscript “K”, right below equation S6.

8. Below Eq. 7, $G_k - 1 < 1$ should be $G_k - 1 < 0$.

We thank the Reviewer for spotting this error, which has been rectified in the updated submission.

RESPONSE TO REVIEWER 2

In general, the results of the paper are very interesting and important and suitable for a wide audience. The manuscript is well written. The figures are clear and illustrative. The authors have made a good effort to support their main claims and to justify their conclusions. They also included very detailed information about their measurements, setup, and characterizations in the supplementary information. But prior to recommending the publication of the manuscript in Nature communications, I would like to ask the authors to address a few comments and questions:

We thank the Reviewer for their appraisal of our work and its broad appeal. Below we address each individual comment of the Reviewer. The comments are highlighted in blue with our responses in black below:

1. The photo of the actual device is very small and buried deep in the supplementary material. In my opinion, the device photo or circuit should be included in the main text (figure 1a is not representative) or at least it should be enlarged in the supplementary information and its various elements should be described in more detail in the text. Also, the circuit symbol of an inductor to ground is confusing. It could refer to the part of the device that implements an inductance to ground or indicate that there is an additional inductor to ground that is not shown. Maybe include an additional drawing which better represents the circuit elements of the layout in a more representative way than the equivalent lumped parallel RLC drawn in c.

We thank the Reviewer for this helpful feedback. While Figure 1a is not the actual device, we feel that it is representative of the key region (the resonator). It represents how on one side we have the pump tone and DC current entering the resonator, and on the other we have the vacuum state entering, being squeezed and reflecting out of the resonator through the same port. Our intention was then to use Fig. S7 to describe the entire device and the role of each part in detail. We have taken the Reviewer’s advice and modified Supplementary Information Figure S7 by i) enlarging panel (b) to show greater device detail, ii) more explicitly indicating in the circuit diagram which element is which by introducing matching colour coding, iii) replacing the RLC equivalent circuit of the resonator with a more accurate LC transmission line, and iv) removing the inductor circuit symbol to avoid insinuating that there is an additional inductance not shown (which there is not). We have significantly extended the figure caption for panel (c) (the circuit diagram) to clearly describe each part of the schematic. Combined with our description of the device in Supplementary Information Section III.B, having an image of the entire design (Fig. S7 a), and a table of all relevant dimensions (Table S1), we hope the readers will appreciate and understand the device design clearly.

2. On several occasions, the authors refer to their resonator as a $\lambda/2$ resonator, is it really a transmission line resonator whose length is $\lambda/2$ (where λ is the resonance wavelength) or is it a lumped element device with a finite number of interdigitated fingers to ground extending from a lumped inductor in the form of a narrow superconducting wire?

The Reviewer asks a very good question regarding whether the resonator can be viewed as lumped elements or is distributed in nature. The correct way to view the resonator is as a lumped-element (or “artificial”) transmission line, with an enhanced inductance and capacitance per unit length. The resonator is a series of repeating unit cells, with each unit cell consisting of the capacitance C to ground of one set of fingers and the inductance L from the thin wire connecting one set of fingers to the next. Each unit cell possesses a characteristic impedance $Z_0 = \sqrt{L/C}$ and a low-pass cutoff $f_c = 1/(\pi\sqrt{LC})$. For frequencies far below the f_c the lumped-element transmission line behaves like a conventional transmission line, as described in Refs. [5,6] of the Supplementary Information. By using the total number of unit cells $N \approx 19$, the resonance frequency $\omega_0/2\pi \approx 6.2$ GHz that we have measured and the simulated impedance of the resonator $Z_0 = 101.7 \Omega$ listed in Table S1, we extract $L = Z_0/(N\omega_0) = 137.4$ pH and $C = 1/(N\omega_0 Z_0) = 13.3$ fF. From this we estimate $f_c = 235.6$ GHz, orders of magnitude above $\omega_0/2\pi$. We therefore conclude that the resonator behaves like a conventional transmission line with impedance Z_0 . We have now included the new Refs. [5,6] and this explanation in Section III.B of the Supplementary Information.

3. Are there other resonance frequencies of this KIPA or higher harmonics? If yes, do they play a role in the device performance?

The Reviewer asks an insightful question about higher harmonics. The resonator should, in theory, have harmonics at $2\omega_0, 3\omega_0, \dots$. The presence of these higher harmonics would indeed impact the device performance, for example, a resonance at $3\omega_0$ would be coupled to the fundamental mode at ω_0 by the strong pump tone at $\omega_p = 2\omega_0$ applied for amplification. This would lead to unwanted competing processes, which would ultimately limit squeezing.

Whilst the higher harmonics do exist, they do not play a role due to the weak anharmonicity introduced by the kinetic inductance nonlinearity and due to the dispersive nature of the of the interdigitated resonator, which detunes the harmonics from integer multiples of ω_0 values. Unfortunately, we were not able to directly probe the higher harmonics in this setup due to the bandwidth of the circulators and filters, though we have observed harmonics and their detuning with other devices. We have discussed the detuning of the harmonics in previous papers, see Refs. [7, 19] of the main text.

We have added the following sentence to the first paragraph of Section II in the main text (lines 89-92): “Here ω_0 is the resonance frequency of the fundamental mode, with higher-order modes detuned away from multiples of ω_0 by the kinetic inductance nonlinearity and dispersion [7, 19].”

4. How was the inductor to ground that sets the coupling Q of the device determined? Through simulations, or measurements of different devices with different inductance values?

We performed both simulations and measurements of iterations of the device design to achieve the target coupling Q , varying the width and length of the shunt inductance. We have added a more detailed description at the end of Supplementary Information Section III.B (lines 214-216) to explain this.

5. In several locations, the authors use the term nondegenerate gain or nondegenerate amplification when maybe the better term is phase-insensitive gain/amplification (since the KIPA has only one mode and one port).

We agree with the Referee that the terms “phase-insensitive/phase-sensistive” are more appropriate in our situation and have updated the manuscript accordingly.

6. Is the 20 dB attenuation of the lowpass filter at the pump frequency enough to block the pump from being transmitted through the device microwave port?

The Referee raises an excellent point about pump tone isolation and whether 20 dB attenuation is sufficient. In fact, prior to the current design we went through two other device iterations with less attenuation in the low pass filter, which were not sufficient to prevent pump crosstalk between the KIPAs. We would like to note that for a pump tone to leak from SQZ to AMP, or vice-versa, it passes two on-chip low-pass filters (~ 40 dB of attenuation) as well as the triple junction circulator (where it lies out of the operational band).

We went to great lengths to check that any residual leakage of the pump tones was not sufficient to impact the squeezing measurements. Whilst some of these measurements were detailed in the original manuscript (Supplementary Information Section V), we have included additional data to support this claim (see Section V.A of the Supplementary Information). We added a new figure (Fig. S15) in Section V.A that shows the AMP

gain as a function of the SQZ pump phase when SQZ is off (by setting the DC current to zero) and when the SQZ pump is turned on at exactly the operation parameters used to achieve maximum squeezing. We observe no significant difference in the AMP gain, indicating that transmission of the pump tone is not a concern.

7. Are the pump powers cited in the caption of figure 2 correspond to the powers at the device port?

Yes, they are referred to the device input. We now explicitly state this in the caption of Fig. 2 with the sentence: **“Increasing opacity corresponds to increasing the pump power at the device input port...”**. We have also added an explanation in the Supplementary Information (see Section I.A, lines 16-20) on how we estimated the line losses, which was used to determine the pump power at the device input.

8. I might have missed it but where in the main text or supplementary information do you cite and discuss the extended data figures?

We thank the Referee for this question. Indeed, one extended data figure was not referenced in the main text. In the original manuscript, we had referenced the following Extended Data figures:

- Extended Data Figure 5 (Thermal squeezing fit): Second-last paragraph of Section V
- Extended Data Figure 6 (Alignment): End of Section IV and in the last paragraph of the section “Homodyne reflection measurements” in Methods.
- Extended Data Figure 7 (Extrapolation): Last paragraph of Section V.

We have now added an explicit reference to all extended data figures, and updated their order in accordance with the order of appearance in the main text. The extended data figures are now referenced in the following locations:

- Extended Data Figure 5 (Frequency and bandwidth dependence): Panel a – end of first paragraph in Section IV. Panel b – third paragraph of the section “Homodyne reflection measurements” in Methods.
- Extended Data Figure 6 (Alignment): End of Section IV and in the last paragraph of the section “Homodyne reflection measurements” in Methods.
- Extended Data Figure 7 (Thermal squeezing fit): Second-last paragraph of Section V.
- Extended Data Figure 8 (Extrapolation): Last paragraph of Section V.

9. The gain-bandwidth of the device is quite small \$\sim 17\$ MHz, could you discuss in the paper how this figure can be increased and how that might affect the device performance?

We thank the Referee for raising this point and allowing us to provide insights on how to increase the gain-bandwidth product (GBP) in future devices. We note that we achieved a GBP that was a factor of ~ 3 larger in a previous device [7], so this is certainly something that can be improved.

In a resonant degenerate parametric amplifier (DPA), it can be shown that the GBP is set by the resonator linewidth $\kappa_L = \omega_0/Q_L$ (where Q_L is the loaded quality factor) [X. Zhou, et al., Physical Review B 89, 214517 (2014)]. Gain in a DPA is realized as the pump strength, equal to $|\xi| = \omega_0 I_{DC} I_p / (4I_*^2)$ for a KIPA (see main text Section VIII and Ref. [7]), is increased towards the instability point $|\xi| = \kappa/2$ [7]. This sets an upper bound for the GBP of $\kappa = 2|\xi|$. In the present devices, we opted for smaller resonator linewidths κ_L (or larger quality factors) compared to previous work in order to reduce the pump current I_p (and therefore pump power) needed to operate the devices. This minimized any pump-induced heating that could have restricted the amount of vacuum squeezing.

To increase the GBP we must raise κ_L , which implies higher pump currents. We believe that we are already operating at the upper limit of the pump powers that can be applied in such delicate experiments as squeezing, thus strategies for increasing I_p without raising the pump power must be adopted. We can increase I_p for a given pump power by lowering the resonator impedance (e.g. by increasing the interdigitated capacitance) [7]. In addition, we can reduce the I_p required by lowering the resonator I_* , which could readily be achieved by decreasing the resonator wire width or the film thickness [25]. Since I_* is proportional to the critical current (I_c) of the device [Zmuidzinas, Annu. Rev. Condens. Matter Phys. 3, 169 (2012)], where in the present study $I_* \approx 5I_c$, then the current upper limit for the pump strength is $|\xi| \approx \omega_0 \times 1/25 \times 1/4 = 2\pi \times 62$ MHz. This sets an upper bound for the GBP of $2\pi \times 124$ MHz, which is sufficient to produce 20 dB of vacuum squeezing over 12.4 MHz of bandwidth. Theoretically $I_* \approx 2.38I_c$ [Zmuidzinas, Annu. Rev. Condens. Matter Phys. 3, 169 (2012)]. I_* could be brought closer to this theoretical limit by eliminating weak spots and current-crowding in the resonator [J. R. Clem and K. K. Berggren, Phys. Rev. B 84, 174510 (2011)], which could boost the GBP

upper limit by more than a factor of 4 to $2\pi \times 547$ MHz. Extending further, one could consider moving away from resonant geometries to travelling wave designs, which has been recently explored in Josephson junction based devices [14].

We have included this detailed discussion on GBP and future improvements in the revised Supplementary Information, Section IV.D.

10. Is the lowpass filter used at the mixing chamber a series RC like the one on the 4 K stage, with the same R and C but with a copper wire? What is the R? Does it cause heating when the dc current is applied?

The filters at the mixing chamber (MXC) are made with copper wire instead of nichrome, specifically to lower the resistance and minimize heating effects. Compared to the 4K filters, the MXC filters have a two order of magnitude lower resistance: $R = 0.4 \Omega$, measured at room temperature. Throughout the experiments, we did not observe any significant temperature changes at the mixing chamber due to the applied DC current. We have added the value for the MXC filter resistance in Section I.A of the Supplementary Information in the third paragraph. Further, the Reviewer’s question helped us to identify an error in our wiring diagram (Supplementary Information Fig. S1) – the MXC low-pass cut off frequency should be 200 kHz, consistent with the lower resistance value.

11. The labels d and e for figure 3 are far above their respective graphs.

We thank the Referee for identifying this issue, which has now been fixed.

12. Given the narrow bandwidth of the KIPAs is it difficult to align their resonance frequencies? How precise is the alignment and how long does it stay stable?

We use the DC current tunability of the KIPAs to align their resonance frequencies, which was not a significant challenge. As we demonstrate in Fig. 2a of the main text, each KIPA is tunable over a > 200 MHz range. We used a voltage source with a resolution of 1 mV to supply the DC current, which converted to a $90 \mu\text{A}$ current resolution at the input of the device. The slope of ω_0 vs. I_{DC} is 220 MHz/A at the optimal point of operation. Therefore, we were able to make frequency steps of 20 kHz, a factor 6 smaller than the integration bandwidth of the squeezing measurements.

In addition, the frequencies and the KIPA gain curves are so stable that we could rely on a look-up table of the frequencies vs DC current values, even between thermal cycles with only slight re-tuning. We have added additional text to Section IV in the main manuscript (lines 200-206) describing the ability to align the frequencies with DC current and perform high levels of squeezing over an extended frequency range. We also put more emphasis on the reproducibility of the gain curves by extending the Supplementary Information Section IV.G and referring to Fig. S11, which depicts for both KIPAs the pump power needed to obtain 20 dB of gain at each frequency operating point over several thermal cycles of the refrigerator.

It was the phase stability of the signal, local oscillator and pump tones that presented the greatest challenge in our experiments and something we accounted for using a re-alignment procedure as described in the last paragraph of the section “Homodyne reflection measurements” in Methods, where we also refer to Extended Data Figure 6.

RESPONSE TO REVIEWER 3

In this paper, the authors thoroughly demonstrate the advantages of a novel 3WM kinetic-inductance parametric amplifier, by showing record level “directly-measured” microwave squeezing of 7.8 dB, resilience to environments with elevated temperature (1K), and external magnetic fields (1T). The record level of squeezing is made possible because kinetic inductance has relatively low spurious 4WM nonlinearity, which is typically a limiting factor of the level of squeezing achievable by a Josephson-junction based amplifier. Resilience to high temperature/magnetic field is made possible by the high band gap of the material of choice (NbTiN). Even though this is not the first time such an amplifier device has been reported, the characterizations that reveal unprecedented squeezing performance are new. The “Squeezes state receiver” setup is very thoughtfully designed and experiments are well executed. The work will inspire the community to explore the use of squeezing in applications that were not possible before. Therefore, it deserves to be published on Nature Communications, if the questions below are addressed.

We thank the Referee for their positive appraisal and useful feedback. In the following we address the technical comments raised. The comments are highlighted in blue with our responses below in black:

1. **Validity of directly-measured squeezing level results:** For the comparison of P_N when squeezer is on vs off to be valid (fig 3b and 3c), an important underlying assumption needs to be true – the system gain (from the output of the squeezer and the room temperature ADC) needs to be the same when the squeezer is turned on and off. While in reality, it is not trivial to maintain consistent KIPA amplifier gain when the input signal changes drastically. For example, by turning on the squeezer, it is certainly possible that the leaked squeezer pump slightly saturates the amplifier, and therefore reduces the gain of the amplifier. In that case, the squeezing level inferred from Figure 3c **COULD BE EXAGGERATED**. Can the author talk about what test they have done to rule out this possibility? (maybe from the view of 1dB compression point?) Results in supplemental material 5 bring some confidence, but are not sufficient. Evidence of consistent amplifier gain when SQZ is on vs off needs to be shown at all conditions where the squeezing level is measured.

The Referee raises a valid point, one has to be extremely careful when performing squeezing measurements to avoid saturation or pump cross talk, which may lead to the squeezing level being over estimated. This was a very serious consideration of ours from the start of our experiments and was what motivated us to perform the cross talk measurements noted by the Reviewer in Fig. S16 of the Supplementary Information. To further support our claim that the AMP gain is the same with the SQZ on or off, we have added two sections in the Supplementary Information (Section IV.F and Section V.A). In Section IV.F, we report the 1 dB compression point power (see Fig. S10) of both KIPAs. We show that the 1 dB compression power at 35 dB of gain, close to the operation point of the AMP where we obtained maximum squeezing (i.e. 34 dB of gain) occurs at -88 dBm (referred to the input of the device), which is substantially larger than the anti-squeezed noise power leaving SQZ. We estimate the anti-squeezed vacuum noise power at the output of SQZ to be $P_{\text{out}}^S \approx (1/4)G_S\hbar\omega_0BW$, equal to -129 dBm for $10\log_{10}(G_S) = 18$ dB (the optimal SQZ gain), orders of magnitude lower than the 1 dB compression power of AMP which confirms that the AMP gain is not saturated by the amplified noise exiting SQZ. Here BW denotes the SQZ bandwidth, \hbar the reduced Planck constant and ω_0 the resonance frequency of SQZ and the 1/2 comes from the number of noise photons of the vacuum state.

In the newly created Section V.A, we have included an additional figure (Fig. S15) that shows the AMP gain as a function of the SQZ pump phase when SQZ is off and when SQZ is on at the operation parameters used to obtain the maximum level of squeezing reported. We believe that it is sufficient to show cross-talk measurements at this operation point, since (i) it is where we claim the -7.8 dB record for vacuum squeezing, and (ii) here the gains are the largest (before the squeezing begins to degrade) and therefore should be most sensitive to the effects of pump leakage. As evident in Fig. S10, no difference in the AMP gain is observed between the two configurations. We believe that this additional data rules out the possibility of exaggerated squeezing due to saturation or pump leakage, addressing the concerns of the Reviewer.

2. **Can the author talk more about the material and design of the magnetic shield used to protect AMP and circulator?** Maybe adding a picture to figure S2 would be helpful. Shielding 2T of field for components that are within a few centimetres seems not a trivial task (distance between circulator and SQZ in figure S2a)

We apologise for causing confusion regarding the relative placement of the different components in our experiments and thank the Referee for highlighting this. Shielding a 2 T field for components separated by a few centimetres would indeed be an extremely challenging task. However, for the magnetic field experiments SQZ was located inside the center of a vector magnet, well away from the AMP and circulator. The circulators and the AMP were shielded with 1 mm thick sheets of lead, which are superconducting below a critical temperature of 7.2 K. We were not attempting to shield the full 2 T field, but rather the stray field from the magnet at the mixing chamber (MXC) plate, which were separated by a distance of 37 cm. The stray field at the MXC was estimated to be 10 mT at a 2 T field in the center of the magnet, as per the Bluefors vector magnet specifications. We describe this in Section IV.H of the Supplementary Information (second paragraph) with a citation [2] to the manual. To clear the confusion we caused in our previous manuscript regarding component placement, we have added a new figure (Fig. S3) and two new paragraphs to the end of Section I.A (lines 29-44) of the Supplementary Information to describe the changes to the setup when performing squeezing in magnetic fields and at elevated temperatures:

“The wiring diagram shown in Fig. S1 illustrates the setup of the squeezing experiments presented in Fig. 3 in the main text. For the squeezing experiments performed in a magnetic field shown in Figs. 4a-c, we modified the setup slightly. As depicted in Fig. S3b, we installed a 6 T/1 T/1 T vector magnet and mounted SQZ in the center of the solenoid, connecting SQZ to the triple junction circulator with a 56 cm long superconducting NbTi coaxial cable. We emphasize that all other wiring and the placement of the AMP and the circulator remain unchanged. The circulator and the AMP were wrapped in 1 mm thick lead sheets (which are superconducting below a critical temperature of 7.2 K) to shield these components from the stray field expected to reach the mixing chamber plate (see Fig. S3a). Based on the specifications of the magnet, we estimate this stray field to

be order 5 mT per 1 T applied in the vertical coil [2]. We note that the lead shielding was installed around the existing setup and therefore could not cover all relevant components entirely, potentially allowing some of the stray field through.

For the squeezing measurements as a function of temperature presented in Figs. 4d-g in the main text, we modified the setup such the SQZ and the triple junction circulator were mounted on the thermal noise source shown in Fig. S2c. A superconducting NbTi coaxial cable connected SQZ and the circulator to avoid creating a thermal link. The thermal noise source was thermally isolated from the mixing chamber plate, as described in Section I.C, which ensured that SQZ and the circulator were thermally isolated from AMP. We again emphasize that that all other wiring and the placement of AMP remain unchanged.”

3. Figure 4d can be made more obvious that the input to the SQZ is also heated. Maybe include a drawing of the 30 dB attenuator at the “input” of the circulator?

This is a great suggestion by the Referee and we have now updated Fig. 4d to include the input 30 dB attenuator. We also added a sentence to the figure caption and in the main text (line 299) to explain that the 30 dB attenuator is thermalized to the heat source and the input state is heated.

4. Typo In supplemental material 3B: “The LP-SIF is introduced to filter out the pump tone at $\omega_p \approx 2\omega_0$ in order to limit crosstalk between SQZ and AMP (see Fig. S6d).” The authors may intend to refer to Fig S6e instead.

We thank the Referee for spotting this error. This has been corrected in the revised manuscript.

REVIEWERS' COMMENTS

Reviewer #1 (Remarks to the Author):

The authors have addressed all of my concerns satisfactorily. I recommend publishing this work in Nature Communications.

Reviewer #2 (Remarks to the Author):

The authors made meaningful changes to their manuscript and adequately answered my questions and the concerns expressed by the other referees. I recommend the publication of the revised version in Nature communications.

Reviewer #3 (Remarks to the Author):

The authors appropriately address most of our concerns. We recommend publication.